# Involvement of a Basic Helix-Loop-Helix Gene *BHLHE40* in Specification of Chicken Retinal Pigment Epithelium

**DOI:** 10.3390/jdb10040045

**Published:** 2022-10-29

**Authors:** Toshiki Kinuhata, Keita Sato, Tetsuya Bando, Taro Mito, Satoru Miyaishi, Tsutomu Nohno, Hideyo Ohuchi

**Affiliations:** 1Department of Cytology and Histology, Okayama University Graduate School of Medicine, Dentistry and Pharmaceutical Sciences, 2-5-1 Shikata-cho, Kita-ku, Okayama 700-8558, Japan; 2Department of Cytology and Histology, Okayama University Faculty of Medicine, Dentistry and Pharmaceutical Sciences, 2-5-1 Shikata-cho, Kita-ku, Okayama 700-8558, Japan; 3Bio-Innovation Research Center, Tokushima University, 2272-2 Ishii, Ishii-cho, Myozai-gun, Tokushima 779-3233, Japan; 4Department of Legal Medicine, Okayama University Faculty of Medicine, Dentistry and Pharmaceutical Sciences, 2-5-1 Shikata-cho, Kita-ku, Okayama 700-8558, Japan; 5Department of Cytology and Histology, Okayama University Medical School, 2-5-1 Shikata-cho, Kita-ku, Okayama 700-8558, Japan

**Keywords:** basic helix-loop-helix e40, *BHLHE40*, LIM homeobox 1, *LHX1*, chicken, optic vesicle, retinal pigment epithelium, RPE, neural retina

## Abstract

The first event of differentiation and morphogenesis in the optic vesicle (OV) is specification of the neural retina (NR) and retinal pigment epithelium (RPE), separating the inner and outer layers of the optic cup, respectively. Here, we focus on a basic helix-loop-helix gene, *BHLHE40*, which has been shown to be expressed by the developing RPE in mice and zebrafish. Firstly, we examined the expression pattern of *BHLHE40* in the developing chicken eye primordia by in situ hybridization. Secondly, *BHLHE40* overexpression was performed with in ovo electroporation and its effects on optic cup morphology and expression of NR and RPE marker genes were examined. Thirdly, we examined the expression pattern of *BHLHE40* in *LHX1*-overexpressed optic cup. *BHLHE40* expression emerged in a subset of cells of the OV at Hamburger and Hamilton stage 14 and became confined to the outer layer of the OV and the ciliary marginal zone of the retina by stage 17. *BHLHE40* overexpression in the prospective NR resulted in ectopic induction of *OTX2* and repression of *VSX2*. Conversely, *BHLHE40* was repressed in the second NR after *LHX1* overexpression. These results suggest that emergence of *BHLHE40* expression in the OV is involved in initial RPE specification and that BHLHE40 plays a role in separation of the early OV domains by maintaining *OTX2* expression and antagonizing an NR developmental program.

## 1. Introduction

The eye develops as bilateral optic vesicles from parts of the forebrain. Subsequently, the optic cup develops by invagination of the optic vesicle. The neural retina (NR) and the retinal pigment epithelium (RPE) differentiate from the inner and outer layers of the optic cup, respectively. The NR gives rise to various neuronal cells involved in photoreception (cone and rod cells) and processing of visual information (interneurons, such as amacrine, horizontal and bipolar cells and projection neurons, such as retinal ganglion cells, and Müllar glia cells). The RPE is a single layered epithelium that prevents light scattering in the retina and sustains metabolism and maintenance of the photoreceptors. How these two retinal domains are initially specified and develop to mature cell layers consisting of various retinal cell types is a fundamental issue in developmental biology and regenerative medicine to alleviate congenital retinal disorders.

In chicken embryos, FGF1 secreted from the surface ectoderm abutting the optic vesicle can direct the neural retina (NR) domain in the optic vesicle [1]. NR cells are specified by stage 13 (19 pairs of somites), and before this stage, optic vesicle cells can exhibit a dorsal telencephalic character when cultured alone. Then, the NR identity is established by BMP signals from the lens ectoderm [2]. However, another study shows that a high dose of BMP5 is able to reprogram NR into the RPE by inducing the expression of *MITF*, an RPE differentiation transcription factor, when a Wnt2b signal is present in the chicken optic cup [3]. This discrepancy is explained by the optic cell behavior depending on the concentration of BMP5 as a low dose of BMP5 induces the expression of *VSX2*, an NR specification transcription factor, resulting in the conversion of the RPE to the NR [3]. Strikingly, embryonic stem cells from mammals can self-organize to form optic vesicles and develop the layered retina even without surface ectoderm [4,5]. Therefore, it seems likely that the NR and RPE compartments are initially determined by intrinsic factors, but their further development and sustained differentiation require extrinsic factors from the surrounding tissues.

Recent studies in zebrafish have shown that NR development is a default state and specification of RPE in the optic primordia separates the distinct NR and RPE domains in the optic cup [6]. However, differences between species are observed in the development of RPE: Maturation of RPE is fulfilled by cell shape change in zebrafish but in amniotes cell proliferation occurs in that phase [7]. In this study, we focus on an RPE gene, *basic helix-loop-helix e40*, (*BHLHE40*), shown to be expressed in the prospective and definitive RPE cells. In zebrafish, *bhlhe40* expression begins in the optic vesicle, becomes confined to its medial layer, the future RPE, and retains in the RPE throughout embryonic development [8]. Since zebrafish *bhlher40* could mark the prospective RPE cells from as early as 12-somite stage [9], how the RPE progenitors expand over the optic cup has been visualized [7,9]. In the developing mouse eye, *Bhlhe40* is expressed in the prospective RPE by embryonic day 11.5 (E11.5) and its expression persists in the RPE at later stages [10]. Bhlhe40, also known as DEC1 [11], Stra13 [12], and SHARP2 [13], has many physiological and pathological functions, such as in cellular differentiation, circadian rhythm, immunity, and cancer [14,15,16]. However, roles of Bhlhe40 in RPE specification has been elusive. As a first step to know the conserved and species-specific systems in early optic vesicle morphogenesis, we examined the expression pattern of *BHLHE40* during chickens’ early eye development and the effects of its overexpression in the optic primordia.

## 2. Materials and Methods

### 2.1. Chicken Embryos

Fertilized White Leghorn chicken embryos were obtained from Goto Furanjyo (Gifu, Japan). Eggs were incubated on their sides in a humidified incubator at 38 °C until the desired embryonic stage. All embryos were staged according to Hamburger and Hamilton [17]. All experiments were performed in accordance with the Recombinant DNA Advisory Committee and the Animal Care and Use Committee, Okayama University, in agreement with relevant guidelines in Japan.

### 2.2. Isolation of Chicken cDNAs and Construction of Expression Vectors

Chicken embryonic heads were dissected from 10 embryos at Hamburger and Hamilton (HH) stages 16–17, 21 and 24 in ice-cold PBS, and RNA was extracted using an RNAqueous-Micro Total RNA Isolation Kit (Thermo Fisher Scientific, Waltham, MA, USA). First strand complementary DNA (cDNA) was synthesized using RNA and SuperScript III Reverse Transcriptase (Thermo Fisher Scientific, Waltham, MA, USA) with random primers. Partial cDNAs for chicken *BHLHE40* and other RPE marker genes were isolated using the cDNA and primers listed in Appendix A by PCR. The amplified cDNAs were cloned into the pGEM-T Easy (Promega, Madison, WI, USA) or pBluescript II KS (+) (Stratagene, La Jolla, CA, USA) vector and used as templates for in vitro transcription. The synthesized RNAs were utilized for in situ hybridization probes. For in ovo electroporation, we obtained cDNA containing the entire coding sequence (cds) for chicken *BHLHE40*, which was further cloned into the expression vector pCAGGS [18].

### 2.3. Cryosectioning

Embryonic heads or whole embryos were fixed in 4% parafolmaldehyde (PFA) in phosphate-buffered saline (PBS) overnight, washed with PBS, cryoprotected in 20% sucrose/PBS and embedded in OCT compound (Tissue-Tek; Sakura Finetek, Tokyo, Japan) and frozen at −80 °C. Sections were cut at 15 μm or 16 μm on a cryostat (Tissue-Tek Polar B; Sakura Finetek Japan, Tokyo, Japan or Leica 1860; Leica Microsystems, Wetzlar, Germany) and mounted on adhesive glass slides (Matsunami Glass, Kishiwada, Japan). The slides were air-dried for 1 h and stored at −20 °C until further analysis.

### 2.4. In Situ Hybridization (ISH)

Whole-mount ISH (WISH) was performed according to Kawaue et al. [19] by the use of a semiautomatic ISH machine (HS-5100; Aloka, Tokyo, Japan) (Figure 1A–D’, E). The ISH on sections was performed according to Sato et al. [20]. Each RNA probe was used at a concentration of 0.67 μg/mL (WISH) or 0.17 μg/mL (sections) in hybridization buffer. Fluorescence ISH using HNPP/Fast Red Fluorescent Detection set (Roche #11758888001) was performed basically according to the kit instructions but without proteinase treatment for subsequent immunofluorescence (Figure 2B–D). ISH experiments were performed at least in triplicate (Figure 1) or in duplicate (other data).

### 2.5. Western Blot Analysis

DNA encoding chicken BHLHE40 (cBHLHE40) tagged with the HA epitope (YPYDVPDYA) at the C-terminus was inserted into the mammalian expression vector pCAGGS. Expression constructs for Δbasic BHLHE40, and one with acidic extension were similarly tagged with HA. The nucleotide sequences were confirmed by Sanger sequencing. The expression of recombinant proteins was assessed by Western blot at least in duplicate as follows. The plasmid DNA was transfected into HEK293T cells (ATCC, Manassas, VA, USA) using polyethyleneimine max (#24765-1; Polysciences, Warrington, PA, USA). An empty pCAGGS vector was also transfected as a negative control. Two days after transfection, the cells were harvested and lysed with RIPA buffer (#08714-04; Nacalai Tesque, Kyoto, Japan) containing 1% SDS. The protein concentration in the lysate was quantified using a Bradford protein assay (5000006JA; Bio-Rad Laboratories, Hercules, CA, USA) after diluting the SDS. The lysate was then mixed with an equal volume of SDS sample buffer (0.1 M Tris-HCl, 2% *w*/*v* SDS, 20% *w*/*v* glycerol, 0.02% *w*/*v* bromophenol blue, 5% *w*/*v* 2-mercaptoethanol, pH 6.8) and heated at 95 °C for 3 min. Ten micrograms of total protein in each sample was separated by SDS-PAGE (12.5% gel) and transferred onto a polyvinylidene difluoride (PVDF) membrane using a wet blotter. Membranes were blocked with Blocking One (#03953-95; Nacalai Tesque, Kyoto, Japan) at RT for 1 h with gentle agitation. The PVDF membrane was then incubated overnight at 4 °C with primary antibodies against the HA epitope (anti-HA High Affinity from rat IgG_1_, 1:2000, #ROAHAHA Roche; Merck, Darmstadt, Germany) or alpha tubulin (alpha Tubulin Monoclonal Antibody DM1A, 1:600, #62204; Thermo Fisher Scientific, Waltham, MA, USA). After washing four times for 10 min with Tris-buffered saline containing 0.1% (*w*/*v*) Tween 20 (TBST), the membrane was incubated for 90 min at RT with a peroxidase-conjugated anti-rat IgG (1:10,000, #SA00001-15; Proteintech, Rosemont, IL, USA) or anti-mouse IgG (1:5000, #7076; Cell Signaling Technology, Danvers, MA, USA) secondary antibody. Primary and secondary antibodies were diluted in IMMUNO SHOT Reagent 1 and 2 (Cosmo Bio, Tokyo, Japan), respectively. After washing four times for 10 min with TBST, immunoreactive bands were visualized by ImmunoStar LD (FIJIFILM Wako Pure Chemical, Osaka, Japan) for the HA epitope or ImmunoStar Zeta (FUJIFILM Wako Pure Chemical, Osaka, Japan) for alpha tubulin. Images of chemiluminescence were captured with a C-DiGit chemiluminescence Western blot scanner (LI-COR Biosciences, Lincoln, NE, USA).

### 2.6. In Ovo Electroporation

In ovo electroporation was performed as previously described [21]. Briefly, chicken embryos were incubated for 36–39 h until HH stage 10. Each expression plasmid in the pCAGGS vector [18] (1 μg/μL of *EGFP* and 5 μg/μL of *LHX1* or *BHLHE40*) with Fast Green FCF (0.1%) (FUJIFILM Wako Pure Chemical, Osaka, Japan) was first injected into the right forebrain vesicle using Nanoject II (Drummond Scientific, Broomall, PA, USA) and the injection volume was 64.4 nL for each embryo. Immediately after DNA injection, the embryos were electroporated using a sharpened tungsten needle (CUY614T-200, and CUY615; Nepa Gene, Ichikawa, Japan) as a cathode and a platinum electrode (CUY611P3-1 and CUY580; Nepa Gene, Ichikawa, Japan) as an anode. Twenty-four hours post electroporation, the embryos were observed under a fluorescence dissection microscope (Leica M165 FC; Leica Microsystems, Wetzlar, Germany) to evaluate their embryonic stages, location and extent of ectopic expression by co-electroporated GFP fluorescence, harvested, fixed in 4% PFA/PBS at 4 °C overnight, washed, and stored in 100% ethanol at −20 °C until further analysis.

### 2.7. Immunofluorescence

Immunofluorescence (IF) was essentially performed according to Kawaue et al. [19]. Briefly, after blocking with 1% Bovine Serum Albumin in PBS containing 0.1% Triton X-100 and anti-HA (1:200, rat IgG; #ROAHAHA Roche; Merck, Darmstadt, Germany) antibody was applied to the sections and incubated at 4 °C overnight. After washing, the sections were incubated with anti-rat IgG conjugated with Alexa Fluor 488 (1:1000; A-11006; Thermo Fisher Scientific) or Alexa Fluor 568 (1:1000; ab175708; Abcam, Cambridge, UK) and Hoechst 33342 (1 μg/mL) at RT for 60 min. Other antibodies used are listed in Appendix A.

### 2.8. Image Capture and Processing

The colored embryos after WISH were photographed with a digital camera system (Leica DFC310). Tissue sections were observed with bright-field or differential interference contrast microscopy (Leica DM5000B) and photographed with a CCD camera system (Nikon DS-Fi1; Nikon, Tokyo, Japan). Brightfield images of in situ hybridization on tissue sections were also captured with an IDS UI-3290SE-C-HQ camera (Imaging Development Systems, Obersulm, Germany) mounted on a Carl Zeiss Axioplan microscope (Carl Zeiss, Oberkochen, Germany). Fluorescent micrographs (Figure 3K,N; Appendix A) were taken with a Leica upright DM5000B microscope equipped with a Nikon DS-Qi1 CCD camera system. Confocal fluorescence images were collected with a Carl Zeiss LSM 780 laser scanning confocal microscope system with 405, 488, 561 and 633-nm laser lines. Brightness and contrast adjustments were performed for some images, and image manipulation was performed using a ZEN 2012 SP1 black edition. Confocal z-stack images were acquired at 0.606-μm intervals for a total depth of 6.66 μm (Figure 2B–D), at 0.705-μm intervals for a total depth of 9.87 μm (Figure 3A), at 0.705-μm intervals for a total depth of 11.99 μm (Figure 3B–D), at 0.658-μm intervals for a total depth of 12.51 μm (Figure 3E) and at 0.634-μm intervals for a total depth of 12.05 μm (Figure 3F–H). The shown images are maximum intensity projections of these confocal z-series.

## 3. Results

### 3.1. Expression Patterns of BHLHE40 in Chickens’ Early Eye Development

*Bhlhe40* was reported to be expressed in the developing RPE of mice and zebrafish [8,12], but none of chicken was described. We therefore examined the expression pattern of *BHLHE40* in chicken embryos by in situ hybridization (ISH) (Figure 1). We found that *BHLHE40* mRNA expression emerged in the dorsal portion of the optic vesicle (OV) toward the dorsal diencephalon at stage 12 (Figure 1C’,C”). At stage 14, distinct signals of *BHLHE40* were observed in the dorsal region of the OV, the prospective RPE (Figure 1D’,D”) [10]. At stage 17, *BHLHE40* was expressed in the outer layer of the optic cup, the nascent RPE and ciliary marginal zone (CMZ) (Figure 1E,E’). At stage 21, *BHLHE40* continued to be expressed in the early-stage RPE and CMZ (Figure 1F). As reported [8], *BHLHE40* was expressed in the midbrain-hindbrain boundary (Figure 1A–D’) and developing pineal gland (Figure 1D’).

### 3.2. Overexpression of BHLHE40 in the OV Drives Early RPE Development

Since *BHLHE40* was expressed in the RPE-lineage of chicken OV, we next examined whether overexpression of *BHLHE40* would drive early RPE development in chicken optic primordia. To discriminate endogenous transcripts for *BHLHE40* from ectopic ones, we constructed an expression plasmid for chicken *BHLHE40* with a human influenza virus hemagglutinin (HA) tag (cBHLHE40-HA) at the C-terminus. We confirmed its expression and the protein size as approximately 45 kDa by Western blot analysis using HEK293T cells (Figure 2A), showing the constructed plasmid vector successfully produced full-length *BHLHE40* proteins.

By in ovo electroporation (EP), we overexpressed *cBHLHE40-HA* in the early optic vesicle. At 40 h post EP, the ectopic expression of *BHLHE40*, HA-tagged BHLHE40 proteins, was verified by immunofluorescence with anti-HA antibody (Figure 2B). Along with a co-electroporated GFP plasmid (Appendix A), cells with ectopic BHLHE40 proteins were visualized all over the OC (Appendix A). We further verified that *BHLHE40* mRNA was overexpressed in the inner layer of the OC, the prospective NR, after electroporation of the *cBHLHE40-HA* plasmid (Figure 2C,D).

Under these experimental conditions, we examined whether the overexpression of *BHLHE40* could influence the morphology of the optic cup and the expression domains of an RPE specification gene, *OTX2* and an NR specification gene, *VSX2*. After overexpression of GFP alone, OTX2 proteins were detected in the outer layer of the optic cup, the prospective RPE, while none of OTX2 was detected in the inner layer, the prospective NR, abutting the lens vesicle (Figure 3A), similar to their normal expression patterns in the optic cup. After *BHLHE40* overexpression, however, OTX2 was detected in the neuroepithelium abutting the lens vesicle (arrowheads in Figure 3C) and continued to localize to the dorsal and ventral epithelium of the OC. Merged view of HA (Figure 3B) and OTX2 (Figure 3C) showed (Figure 3D) that OTX2 protein in the inner layer of the optic cup abutting the lens vesicle was ectopically induced or its early expression in the optic vesicle was maintained under the influence of *BHLHE40* overexpression. Regarding *VSX2,* its expression was downregulated in that portion (arrowheads in Figure 3G,H), while it was observed in the ventral portion of the inner layer of OC (below the arrowheads).

After one and half day post electroporation, as shown by the normal optic cup on the contralateral control side, the outer layer of the OC remains a single layer of *OTX2*-expressing cells (Figure 3I), while the neuroepithelial cells in the inner layer of the OC proliferates to become a thickened NR expressing *VSX2* by this stage (Figure 3L). After *BHLHE40* overexpression, however, the dorsal portion of the neuroepithelium abutting the lens vesicle remained thin (arrowheads in Figure 3J,M), where *BHLHE40* was overexpressed judging from the localization of co-electroporated GFP (Figure 3K,N). At this stage, *OTX2* mRNA was detected in the neuroepithelium abutting the lens vesicle (arrowheads in Figure 3J), while *VSX2* expression was downregulated in that portion (arrowheads in Figure 3M).

In another embryo (#22), the OV development was disrupted and no lens vesicle formation was observed (Appendix A). Although *OTX2* is normally expressed in the dorsal portion of the OV, the prospective RPE, it was expressed in the distal portion of the deformed OV (Appendix A) where *BHLHE40* was overexpressed, judging from the localization of co-electroporated GFP (Appendix A). Regarding *VSX2*, its expression was reduced but present partly (Appendix A). We further examined whether *BHLHE40-HA* overexpression might induce cell death by immunostaining with anti-cleaved caspase 3 antibody. At one day post electroporation, we did not find any positive cells for cleaved caspase 3 in the HA-immunoreactive cells (Appendix A). These data suggest that BHLHE40 could maintain *OTX2* expression in the optic vesicle, drive early RPE development and repress the developmental program for NR.

### 3.3. Effects of LHX1 Overexpression on BHLHE40 Expression

BHLHE40 lacking a basic region (Δbasic BHLHE40) is known to act as a dominant negative form of BHLHE40 [14]. Therefore, we next examined whether overexpression of Δbasic *BHLHE40* could alter the morphology of the optic cup, such as lack of RPE development or formation of double NR. We constructed an expression plasmid for Δbasic BHLHE40 with an HA tag, whose calculated molecular mass was 42,877 Da and confirmed its expression by Western blot analysis (Appendix A). We overexpressed Δbasic *BHLHE40* in the early optic vesicle. However, the eye morphology was not altered (Appendix A and not shown). Additionally, an extension of 12 acidic amino acids in place of the basic region of bHLH transcription factors is known to stabilize heterodimerization between the dominant negative forms and endogenous bHLH proteins [22]. Therefore, we further constructed an acidic extension form of Δbasic *BHLHE40* and expressed it in the optic vesicle. However, despite confirmed protein expression for the acidic extension form (39,998 Da with HA) (Appendix A), overexpression of this construct did not alter the eye morphology, either (Appendix A and not shown).

We therefore sought to utilize the *LHX1* overexpression. Previously, we found that *LHX1* overexpression in the early chicken OV induced the second NR formation, verified by ectopic expression of NR marker genes, *RX, SIX3, SOX2, SIX6,* and *VSX2,* and called double NR [19]. Recent studies in zebrafish show that the default state of neuroepithelium of the OV is the NR, and expression of RPE specification genes drive the development of RPE and separates the NR and RPE domains [6]. In chicken embryo, however, deducing from overlapping expression domains of the above NR genes and RPE genes. such as *OTX2* and *MITF*, suggests bipotential retinal progenitor cells in the nascent OV. Furthermore, *LHX1* expression in the posterior portion of the chicken OV corresponds to the future central NR domain [23], suggesting its ability to drive NR formation by antagonizing RPE differentiation. Thus, we examined whether *BHLHE40* expression was downregulated in the *LHX1*-oversexpressed optic cup. We analyzed it in chicken embryos at 24 h post electroporation of approximately stages 15 to 16, by ISH (Figure 4). After *LHX1* overexpression, protruding regions were observed from the outer layer of the optic cup, where *OTX2* and *MITF*, the RPE genes are repressed [19]. Sectioning of the *LHX1*-overexpressed embryonic heads and ISH on sections revealed that *BHLHE40* expression was downregulated in the portion transforming to NR from the nascent RPE in the outer layer of the OC (arrows in Figure 4C,D and Appendix A). Thus, *BHLHE40* expression is negatively regulated by a molecular force of NR formation, *LHX1*.

## 4. Discussion

A bHLH transcription factor gene *Bhlhe40* was reported to be expressed in the developing RPE as well as somites and other organ primordia of mice and zebrafish [8,12]. Recently, this gene has been used to visualize the cell behavior during early RPE development [7,9]. Here, we firstly described the early onset of *BHLHE40* expression in the dorsal region of the optic vesicle, the prospective RPE of the chicken [10]. While we focused on the role in RPE development, it has been known that BHLHE40 is involved in circadian rhythm and tumor progression [24]. There is a paralogous gene of *BHLHE40*, *BHLHE41*, which is also expressed in the developing RPE of zebrafish [6], chicken (this study, Appendix A), and human iPS cell-derived RPE [25]. In human tumor cells, there are functional differences in their roles: BHLHE40 induces apoptosis and epithelial-mesenchymal transition (EMT), whereas BHLHE41 inhibits them [26]. This is considered to be due to their structural differences in which both of BHLHE40/41 have bHLH and Orange domains, but BHLHE41 has an additional alanine/glycine-rich region in the C-terminus [26] (Appendix A). Regarding this, although there are similar upstream regulators for the two paralogs, their downstream effectors and influences are rather opposite. For example, BHLHE40 negatively regulates E-cadherin and Bcl-2, while BHLHE41 negatively regulates Slug (Snail2), Twist1, and caspase-8 [26]. In addition to suppression of these target genes by binding to E-boxes, BHLHE40/41 regulate each other by binding to their E-boxes. Although we have yet to examine the effect of *BHLHE41* overexpression, the involvement of BHLHE41 might be correlated to the result of no phenotypes when dominant negative (DN) forms of *BHLHE40* were overexpressed, as overexpression of the DN forms might induce some compensation by the paralog.

Not a few downstream targets of BHLHE40 have been identified in other contexts [16]. Along with EMT-related genes, BHLHE40 represses Cyclin D1 and STAT1 and upregulates beta-catenin by cooperating with other co-transcription factors. When acting as a repressor, BHLHE40 is known to recruit histon deacetylase, HDAC1. These known target genes represent its multi-functionality and studies on the candidate target genes and transcriptome analysis of *BHLHE40*-overexpressed embryonic eyes will further illuminate detailed molecular aspects of its functions in the chicken eye development.

As we mentioned in the Introduction, *Bhlhe40* was originally isolated as a retinoic acid-inducible gene, *Stra13* [12], and they described that P19 cell clones expressing higher amount of STRA13 died after the passage in culture. In our electroporation method, we did not observe any cell death in *BHLHE40-HA* expressing chicken optic cup cells, as shown in Appendix A. Boudjelal et al. [12] also said that the low expressing cells did not undergo efficient neuronal differentiation, and in this study we could not determine the expression level of *BHLHE40* that robustly altered the morphology and expression of *OTX2* and *VSX2*, which would be our next challenge.

One of the upstream factors of BHLHE40/41 is hypoxia, which is often molecularly exemplified by HIF1a [26]. RPE cell-specific knockout mice of this pathway genes show that regulation of *Hif1a* in the RPE is required for normal RPE and related ocular development [27]. Therefore, during chicken RPE development, it would be intriguing to know the effect of *HIF1A*-hypoxia pathway genes on the expression *BHLHE40*, which will be researched in the next study. Another important upstream regulator of BHLHE40/41 in mouse tumor cells is TGF beta [26,27]. It has been reported that TGF beta member activin from the extraocular mesenchyme surrounding the optic primordia patterns the optic vesicle in the chicken [28]. It is therefore conceivable that the expression of *BHLHE40* would be induced by TGF beta family signals from the extraocular mesenchyme at the early phase of optic cup development.

In zebrafish, RPE progenitors expand around the NR in two phases; firstly RPE progenitors are specified with an increase in cell number and then the RPE stretches around the NR with an elongation of RPE progenitor cells [9]. In amniotes (chicken and human), however, in the second phase, the RPE still expands by cell proliferation with a less pronounced cell flattering [7]. Correspondingly, a reported transcriptome analysis has revealed two waves of transcription factor genes regulating the identity of the RPE [6]: Early peaking genes include *tead3b*, *tfap2a*, and *tfap2c*, and representative late peaking genes are *klf5a* and *klf2b* as well as *otx2* and *bhlhe40*. According to this, we cloned the cDNAs and firstly examined the expression patterns of *TEAD3*, *TFAP2A, TFAP2C, KLF2,* and *KLF5* in chicken embryos, but none of the genes except *TEAD3* were expressed in the developing RPE (Appendix A and Appendix A). Only *TEAD3* was found to be expressed in the developing chicken RPE but its expression domains were not restricted to RPE progenitors as seen for *BHLHE40*. Thus, we suggest that the molecular network regulating RPE specification may not be identical between zebrafish and chicken, except for established RPE genes, such as *OTX2* and *BHLHE40*.

This study showed *BHLHE40* overexpression upregulated *OTX2* expression and downregulated *VSX2* expression, suggesting *BHLHE40* would drive RPE specification in bipotential retinal progenitor cells of the chicken. Since it was reported that *OTX2* alone could not induce the expression of *MITF,* a melanogenic and differentiation factor for RPE in the chicken embryo [29], BHLHE40 alone might not procced with later RPE development in the chicken. In this regard, *MITF* was found to be negatively regulated by *VSX2* in the developing chicken eye [30]. Furthermore, direct repression of *Mitf* by Vsx2 have been reported in mouse [31]. As *VSX2* was markedly repressed in the optic primordia where *BHLHE40* was ectopically expressed, we think that *BHLHE40* positively regulates *MITF* at least through repression of *VSX*2. Additionally, RNA levels of *Cyclin D1* are significantly decreased in the *Vsx2*-deficient *or^J^* mouse [32]. Since Cyclin D1 is known to promote cell proliferation [33], repression of *Cyclin D1* would have disrupted expansion of the prospective NR layer after *BHLHE40* overexpression and also via downregulation of *VSX2.* In cancer cell lines, BHLHE40 (DEC1) interacts with Cyclin E and stabilizes it, leading to prolonging the S phase and suppressing cell proliferation [34]. Thus, a direct negative effect of BHLHE40 on the cell cycle of nascent OV cells would be conceivable as well.

We did not succeed in the functional inhibition of *BHLHE40* in the developing chicken eye to see if the presumptive RPE could transdifferentiate into the NR in the stage of bipotential retinal progenitors. Instead, we utilized *LHX1* overexpression and found that *BHLHE40* is repressed in the transdifferentiating outer layer of the optic cup. Previously, we reported that the second NR formation by *LHX1* overexpression was not mediated by ectopic induction of *FGF8* expression in the early phase [19]. A recent work using fate mapping in chicken embryos has shown that the central NR is defined within the posterior optic vesicle (OV), while the peripheral retina, including future CMZ actually comes from the distal OV [23]. They also showed *LHX1* expression in the posterior portion of the OV corresponds to the future central NR domain. Accordingly, *LHX1* would antagonistically regulate *BHLHE40* in the very early stage of optic vesicle development and determine the central NR domain.

In conclusion, a bHLH gene, *BHLHE40* is expressed in the prospective and definitive RPE cells as well as CMZ of the developing chicken eye. BHLHE40 can influence the expression of *OTX2* and *VSX2* and its expression can be downregulated by an NR-inducing factor LHX1, suggesting its definite role in the NR/RPE developmental programs of the optic vesicle. Conserved expression of *BHLHE40* in the RPE lineage across vertebrates, and its effects on the RPE/NR specification genes verify that this RPE gene belongs to the transcriptional node group among diversified RPE developmental processes.

## Figures and Tables

**Figure 1 jdb-10-00045-f001:**
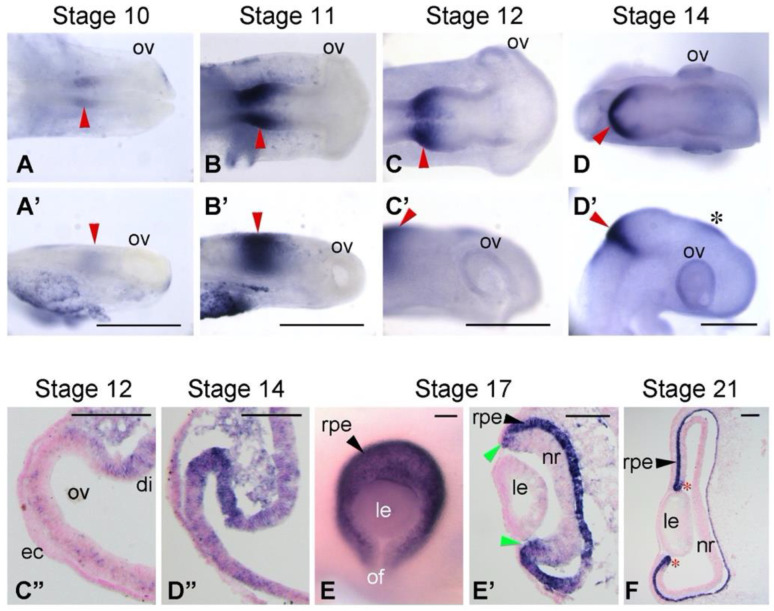
Expression pattern of *BHLHE40* in chicken early embryos. (**A**–**D’**,**E**) Whole-mount in situ hybridization of chick embryos at Hamburger and Hamilton (HH) stages 10 to 14, and 17. Dorsal views (**A**–**D**) and lateral views (**A’**–**D’**; to the same scale of **A**–**D**, respectively) of embryonic heads or eye (**E**) are shown. Red arrowheads show the midbrain-hindbrain boundary. Asterisk in (**D’**) shows the developing pineal gland. (**C”**,**D”**,**E’**,**F**) Frontal sections of the optic primordia at stages 12 to 21. Lateral is to the left. Dorsal is to the top. Light green arrowheads in (**E’**) and asterisks in (**F**) show the ciliary marginal zone. mRNA signals are in dark blue. For sections, cell nuclei were stained with nuclear fast red. di, diencephalon; ec, surface ectoderm; le, lens vesicle; nr, neural retina; of, optic fissure; ov, optic vesicle; rpe, developing RPE. Scale bars: 0.5 mm in (**A**–**D’**), 100 μm (**C”**,**D”**,**E**,**E’**,**F**).

**Figure 2 jdb-10-00045-f002:**
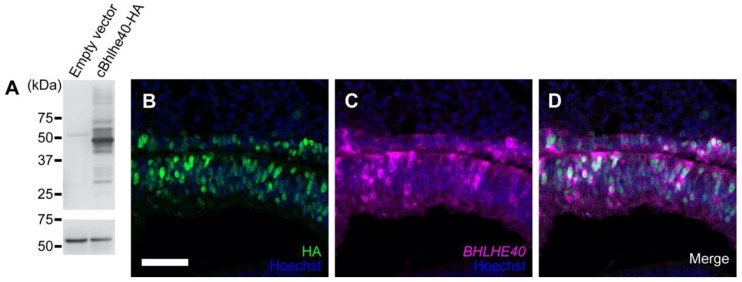
Validation of HA-tagged *BHLHE40* expression. (**A**) Whole cell extracts of HEK293T cells transfected with chicken *BHLHE40-HA*/pCAGGS or empty pCAGGS vector were analyzed by Western blot. HA-tagged proteins were detected by anti-HA antibody. The predicted molecular mass of BHLHE40 is 44,830 Da. Alpha tubulin from the same samples was detected for loading reference. (**B**–**D**) Section of the developing RPE (upper) and NR (lower) layers at 1.5 days post electroporation (HA-tagged *BHLHE40* overexpression). (**B**) Immunofluorescence of HA, showing cell nuclei that express BHLHE40-HA proteins in both retinal layers. Secondary antibody to anti-HA was Alexa Fluor 488-conjugated anti-rat IgG. (**C**) In situ hybridization of *BHLHE40*, showing cells that express *BHLHE40* mRNA in both retinal layers. Color development was performed with HNPP/FastRed. (**D**) Merged image of (**B**) and (**C**). Cell nuclei were stained with Hoechst 33342. Scale bar: 50 μm in (**B**) for (**B**–**D**).

**Figure 3 jdb-10-00045-f003:**
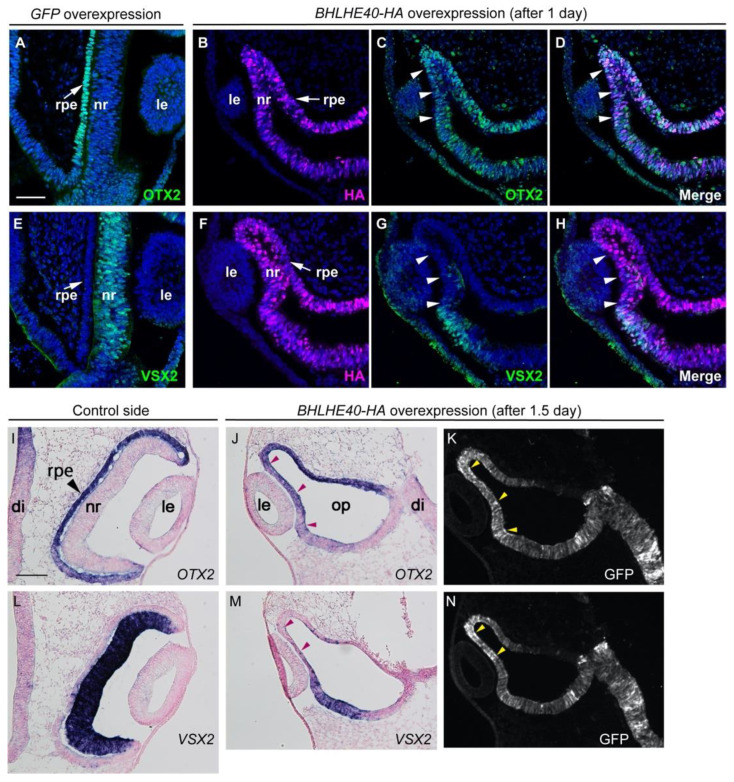
Expression pattern of *OTX2* and *VSX2* in the optic cup after *GFP* and *BHLHE40* overexpression. Frontal sections of the optic cup from three embryos (*GFP*#2, *BHLHE40*#14, *BHLHE40*#17) are shown. Dorsal is to the up. (**A**–**H**) Optic cups at 1 day post electroporation (for *GFP* alone and HA-tagged *BHLHE40* overexpression). (**A**,**E**) Immunofluorescence of OTX2 (**A**) and VSX2 (**E**) in *GFP* overexpression control (#2). (**B**–**D**) Immunofluorescence of HA (**B**) and OTX2 (**C**) in HA-tagged *BHLHE40* overexpression (#14). Secondary antibodies to anti-HA and anti-OTX2 were Alexa Fluor 568-conjugated anti-rat IgG and Alexa Fluor 647-conjugated anti-rabbit IgG, respectively. (**D**) Merged image of (**B**) and (**C**). (**F**–**H**) Immunofluorescence of HA (**F**) and VSX2 (**G**) in HA-tagged *BHLHE40* overexpression. Secondary antibodies to anti-HA and anti-VSX2 were Alexa Fluor 568-conjugated anti-rat IgG and Alexa Fluor 488-conjugated anti-sheep IgG, respectively. (**H**) Merged image of (**F**,**G**). Cell nuclei were stained with Hoechst 33342 (**A**–**H**). Similar data from another embryo (#11) are shown in Appendix A. (**I**–**N**) Optic cups at 1.5 days post electroporation (HA-tagged *BHLHE40* overexpression, #17). (**I**,**L**) Optic cups on the control side are shown. *OTX2* is expressed in the outer layer of the optic cup (**I**), while *VSX2* is expressed in the inner layer of the optic cup (**L**). In (**K**,**N**), co-electroporated GFP, showing the domain of *BHLHE40* overexpression. Images for GFP were taken before ISH. (**J**,**M**) In embryo #17, the inner layer of the optic cup abutting the lens, where *BHLHE40* is ectopically expressed (arrowheads in **J**), remains thin and expresses *OTX2* (**M**). In the same domain, *VSX2* expression is downregulated (arrowheads in **M**). Cell nuclei were stained with nuclear fast red (**I**–**N**). di, diencephalon; le, lens vesicle; nr, neural retina; op, optic cup or deformed optic vesicle in (**G**); rpe, developing RPE. Scale bar: 50 μm in (**A**) for (**A**–**H**), 100 μm in (**I**) for (**I**–**N**).

**Figure 4 jdb-10-00045-f004:**
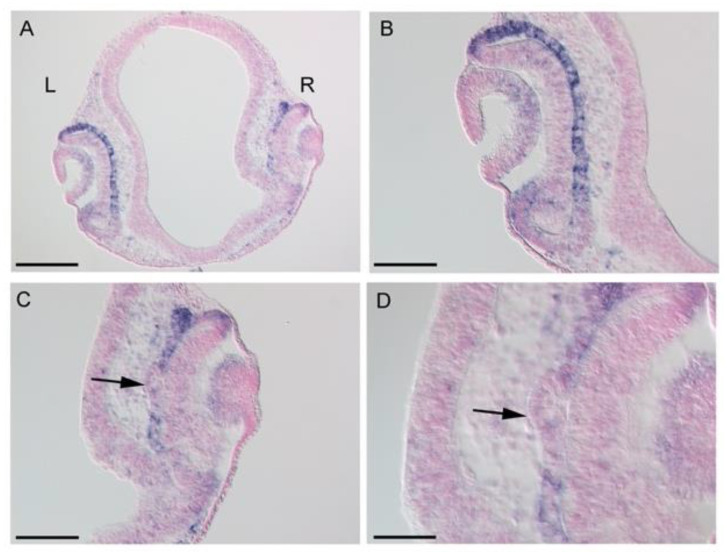
*BHLHE40* expression after *LHX1* overexpression. (**A**) In situ hybridization (ISH) on the frontal section of the chicken embryonic head after 1 day post electroporation (embryo #5). Enlarged view of the left (L) control optic cup (OC) is shown in (**B**), and that of the right (R) *Lhx1*-overexpressed one is shown in (**C**,**D**). *BHLHE40* expression is downregulated in a portion of the outer layer of the OC (arrows in **C**,**D**). Cell nuclei were stained with nuclear fast red. Scale bars: 200 μm (**A**), 100 μm (**B**,**C**), 50 μm (**D**). Similar data from other embryos are shown in Appendix A.

## Data Availability

Not applicable.

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
