# Peer review of "Involvement of a Basic Helix-Loop-Helix Gene BHLHE40 in Specification of Chicken Retinal Pigment Epithelium"

_jdb, 2022, doi:10.3390/jdb10040045_

Round 1
Reviewer 1 Report
This study investigates the expression and function of the bHLH transcription factor bhlhe40 during RPE development in the chick embryo. Bhlhe40 expression starts at HH12 in the presumptive RPE domain of the optic vesicle and is robust from HH14 onwards. Overexpression of bhlhe40 in the optic vesicle at HH10 results in incomplete invagination of the vesicle or small vesicle-like structures. The neural retina transcription factor Vsx2 is not expressed in the electroporated domain while Otx2 expression is maintained. To determine further a potential role in RPE development, Lhx1 was misexpressed in the presumptive RPE causing transdifferentiation into retina with bhlhe40 being downregulated like other RPE genes.
Some further information would greatly enhance the current study:
1. It should be explained in more detail in the introduction what is known about bhlhe40 in zebrafish and mouse (onset and location, functional role).
2. Animal and experiment numbers should be included. There are VSX2 and OTX2 antibodies that could work in chick at these early stages (for example, see PMID 32192535). This would allow to show co-labeling with HA and GFP at a single cell level.
3. It should be shown that bhlhe40 is expressed exactly where HA labeling is found.
4. The current data does not support sufficiently the conclusion that bhlhe40 is involved in initiation of RPE genes. First, bhlhe40 expression starts later than Mitf in the chick optic vesicle (see Suppl. Fig.3A in PMID 19570849). Generally, the expression pattern of the RPE genes Mitf and Otx2 suggest that the RPE fate could be a default state. They are co-expressed with early ocular identity genes, for example, Rx1 or Six3, while more specific and restricted neural retina markers such as Vsx2 are later robustly expressed (HH14). Thus, it is possible that bhlhe40 electroporation into the optic vesicle at HH10 helps to maintain the default RPE state rather than transdifferentiating the retina into RPE. It would therefore be helpful to know whether bhlhe40 electroporation at a stage when the retina is already specified and expresses Vsx2 (HH14) leads to transdifferentiation, which should be also examined using Mitf as a marker.
5. Second, given a potential role of bhlhe40 in modulating cell death, survival (and proliferation) of the electroporated region in the distal optic vesicle should be quantitatively analyzed, preferably before completion of the phenotype at HH17.
7. The data shown in Figure 4 is not very clear, it is difficult to see a difference between electroporated (Figure 4A?) and control eye (Figure 4C?), and the effect of Lhx1 overexpression appears weak in comparison to the electroporation control GFP. Figure 4H needs arrows and more proof for transdifferentiation should be shown.
Author Response
Responses to “Comments and Suggestions for Authors by Reviewer 1”
This study investigates the expression and function of the bHLH transcription factor bhlhe40 during RPE development in the chick embryo. Bhlhe40 expression starts at HH12 in the presumptive RPE domain of the optic vesicle and is robust from HH14 onwards. Overexpression of bhlhe40 in the optic vesicle at HH10 results in incomplete invagination of the vesicle or small vesicle-like structures. The neural retina transcription factor Vsx2 is not expressed in the electroporated domain while Otx2 expression is maintained. To determine further a potential role in RPE development, Lhx1 was misexpressed in the presumptive RPE causing transdifferentiation into retina with bhlhe40 being downregulated like other RPE genes.
Some further information would greatly enhance the current study:
- It should be explained in more detail in the introduction what is known about bhlhe40 in zebrafish and mouse (onset and location, functional role).
Response 1: As suggested by the reviewer, we have explained about bhlhe40 in zebrafish and mouse in more detail in the introduction (page 2, lines 71-79).
- Animal and experiment numbers should be included.
There are VSX2 and OTX2 antibodies that could work in chick at these early stages (for example, see PMID 32192535). This would allow to show co-labeling with HA and GFP at a single cell level.
Response 2: Animal numbers examined have been indicated in each figure or figure legend. Experiment numbers have been described (page 3, lines 119-120). 
We tried two kinds of anti-VSX2 antibodies. One used in PMID 32192535 did work after antigen retrieval in our experiment system. Co-labeling with HA and GFP has been shown in supplementary Figure A1.
- It should be shown that bhlhe40 is expressed exactly where HA labeling is found.
Response 3: We have shown that BHLHE40 is expressed where HA labeling is found in the revised Figure 2B-D.
- The current data does not support sufficiently the conclusion that bhlhe40 is involved in initiation of RPE genes. First, bhlhe40 expression starts later than Mitf in the chick optic vesicle (see Suppl. Fig.3A in PMID 19570849). Generally, the expression pattern of the RPE genes Mitf and Otx2 suggest that the RPE fate could be a default state. They are co-expressed with early ocular identity genes, for example, Rx1 or Six3, while more specific and restricted neural retina markers such as Vsx2 are later robustly expressed (HH14). Thus, it is possible that bhlhe40 electroporation into the optic vesicle at HH10 helps to maintain the default RPE state rather than transdifferentiating the retina into RPE. It would therefore be helpful to know whether bhlhe40 electroporation at a stage when the retina is already specified and expresses Vsx2 (HH14) leads to transdifferentiation, which should be also examined using Mitf as a marker.
Response 4: We do not say BHLHE40 initiates expression of RPE genes, for example, MITF, whose expression is already found at stage 10-11 (Ishii et al., Development, 2009). Since NR and RPE genes are initially co-expressed in the optic vesicle of chicken embryos, the default NR state may be true for zebrafish, but not for chicken. Bipotential state may be suited for the chicken optic vesicle. So, in the revised manuscript, when “the default NR stage” is referred, we have added “in zebrafish” (page 2, line 65; page 8, lines 320).
As for MITF, under our present in situ hybridization protocol [reference #20, PMID: 33427299], we could not obtain MITF mRNA signals at a desired signal-to-noise ratio as shown right.
Since it was the same probe as described in our previous report (reference #19, Kawaue et al., 2012), we should have optimized probe concentration or else. Instead, we tried an anti-MITF antibody (Bio Academia, Japan, Catalog number 73-107, newly purchased, which had been used in Tsukiji N et al: Mitf functions as an in ovo regulator for cell differentiation and proliferation during development of the chick RPE. Dev Biol 326: 335-346 (2009), PMID: 19100253), but it did not work unfortunately. So, here we only examined the expression of OTX2 as an RPE marker. As described in Discussion and shown in Figures A9-11, we also attempted to find new RPE markers that could be used in the developing chicken eye in vain.
- Second, given a potential role of bhlhe40 in modulating cell death, survival (and proliferation) of the electroporated region in the distal optic vesicle should be quantitatively analyzed, preferably before completion of the phenotype at HH17.
Response 5: We examined cell death in the optic vesicle 24 hours post electroporation (around HH stage 15) by immunostaining for activated caspase 3. As shown in the new Supplementary Figure A3, cell death was not detected in the BHLHE40-HA overexpressed cells in the optic cup. As for cell proliferation, we examined it using our available anti-Ki-67 antibody (SP6), but it did not work unfortunately. We should try other antibodies or methods to examine cell proliferation, but we would like to do that in the next study. The possibility of decreased cell proliferation after BHLHE40 overexpression is touched upon in Discussion, though it is via repression of VSX2 (page 10, line 417 to page 11, line 422).
- The data shown in Figure 4 is not very clear, it is difficult to see a difference between electroporated (Figure 4A?) and control eye (Figure 4C?), and the effect of Lhx1 overexpression appears weak in comparison to the electroporation control GFP. Figure 4H needs arrows and more proof for transdifferentiation should be shown.
Response: We have replaced it with the new data for Figure 4 and Supplementary Figure A5: At 24 hours post LHX1 overexpression, BHLHE40 expression in the outer layer of the optic cup disappeared, as indicated by arrows in Figure 4C and D. Molecular property of the optic cup after LHX1 overexpression has been reported previously (reference #19, Kawaue et al., 2012), and we have cited this paper (page 8, line 318-320; page 9, lines 339-340).

Reviewer 2 Report
Manuscript “Involvement of a basic helix-loop-helix gene BHLHE40 in specification of chicken retinal pigment epithelium” by Toshiki Kinuhata and colleagues characterizes the implications of the transcription factor BHLHE40 in the early specification of the neuronal retina (NR) and retinal pigment epithelium (RPE) territories. First, authors investigate the expression patterns of the transcription factor looking at mRNA in situ at various stages during the chick neuronal development. Then, authors investigate the effect of overexpression of this gene in the specification of the eye structures, either of the full-length version, or of a truncated version known to act as a dominant negative. Authors show that overexpression of the full-length version of the BHLHE40 results in ectopic expression of OTX2 and inhibition of VSX2, suggesting a role in the specification of the RPE and NR retina territories. Lastly, authors also use an LHX1 overexpression at similar stages and showed that BHLHE40 expression was reduced compared to the controls, suggesting that LHX1 could regulate BHLHE40 at the incipient stages of optic vesicle development.
The manuscript is written clearly and succinctly and allows a clear analysis of the results. Authors used a combination between immunofluorescence and in situ hybridization to analyze the phenotypes upon various overexpression scenarios, with the predominant use of the latter technique.
Major point – one of the main results of the manuscript is the analysis of the OTX2 and VSX2 specificity, as a read-out for the determination of the RPE and NR territories. The manuscript could be greatly enhanced if, instead of showing parallel expression patterns of the two markers by ISH, authors would use antibody staining to detect both gene products simultaneously. There are commercial antibodies validated in chick at the stages of interest for both OTX2, VSX2, as well as for GFP. Using an antibody to amplify the GFP expression will not only ensure a better signal for the electroporation control, but would also allow the visualization of these factors precisely in the regions electroporated, by doing immunofluorescence of all 3 channels simultaneously. This strategy will overcome the limitation presented in figure 3, where the electroporation control is analyzed prior to ISH, hence providing slightly distorted imaging fields due to subsequent imaging sessions, which cannot be perfectly overlapped (in figure 3, F does not seem to be the electroporation control of the D, considering the appearance of the shape of the layers).
In addition, the manuscript would be enhanced if authors will include a figure presenting the results of the first part of the “3.3 Effects of LHX1 Overexpression on BHLHE40 Expression”, where they overexpressed truncated versions of the BHLHE40 transcription factor.
Lastly, authors should discus further the implications of the discovery of this factor and its expression in the developing eye and give a future perspective about how would these discoveries advance our understanding of the specification of the vertebrate retina and RPE.
Minor points:
Line 42-43: There is no mention of bipolar cells and Muller glia in the initial description of the retina, which provides an incomplete description of the structure.
Line 58: “Strikingly, in mammals, embryonic stem cells can self-organize” – using an expression such as “embryonic stem cells from mammals” would provide more accuracy about these protocols.
Line 112: “sequences were checked by Sanger sequencing” – using “sequences were confirmed by Sanger sequencing” would be preferably in scientific writing.
Line 138: “In Ovo Electroporation” – the word “in ovo” should be italic font, and not “electroporation”.
250-251: “These data indicate that BHLHE40 can drive early RPE development and repress the developmental default state of NR” – to support this data, embryos should be allowed to develop past E5, where the RPE is pigmented and specified.
275 – “After LHX1overexpression” – a space is needed between LHX1 and overexpression
Figure 3 A, B, C, D, E, F seem to be imaged at a different scale. Embryo number should be labeled in the figure not only on text.
Figures 3 and 4 are missing scale bars.
Figure 4 – E, F, G, E’ don’t show nuclear marker, unlike H
Author Response
Responses to “Comments and Suggestions for Authors by Reviewer 2”
Manuscript “Involvement of a basic helix-loop-helix gene BHLHE40 in specification of chicken retinal pigment epithelium” by Toshiki Kinuhata and colleagues characterizes the implications of the transcription factor BHLHE40 in the early specification of the neuronal retina (NR) and retinal pigment epithelium (RPE) territories. First, authors investigate the expression patterns of the transcription factor looking at mRNA in situ at various stages during the chick neuronal development. Then, authors investigate the effect of overexpression of this gene in the specification of the eye structures, either of the full-length version, or of a truncated version known to act as a dominant negative. Authors show that overexpression of the full-length version of the BHLHE40 results in ectopic expression of OTX2 and inhibition of VSX2, suggesting a role in the specification of the RPE and NR retina territories. Lastly, authors also use an LHX1 overexpression at similar stages and showed that BHLHE40 expression was reduced compared to the controls, suggesting that LHX1 could regulate BHLHE40 at the incipient stages of optic vesicle development.
The manuscript is written clearly and succinctly and allows a clear analysis of the results. Authors used a combination between immunofluorescence and in situ hybridization to analyze the phenotypes upon various overexpression scenarios, with the predominant use of the latter technique.
Major point – one of the main results of the manuscript is the analysis of the OTX2 and VSX2 specificity, as a read-out for the determination of the RPE and NR territories. The manuscript could be greatly enhanced if, instead of showing parallel expression patterns of the two markers by ISH, authors would use antibody staining to detect both gene products simultaneously. There are commercial antibodies validated in chick at the stages of interest for both OTX2, VSX2, as well as for GFP. Using an antibody to amplify the GFP expression will not only ensure a better signal for the electroporation control, but would also allow the visualization of these factors precisely in the regions electroporated, by doing immunofluorescence of all 3 channels simultaneously. This strategy will overcome the limitation presented in figure 3, where the electroporation control is analyzed prior to ISH, hence providing slightly distorted imaging fields due to subsequent imaging sessions, which cannot be perfectly overlapped (in figure 3, F does not seem to be the electroporation control of the D, considering the appearance of the shape of the layers).
Response: In our experimental conditions, anti-VSX2 antibody needed antigen retrieval, but anti-OTX2 did not as indicated in Supplementary Table A2. So, unfortunately in this revision, we could not realize double immunostaining for OTX2 and VSX2. But we could perform double staining for OTX2 and HA (ectopic BHLHE40), or VSX2 and HA as shown in the revised Figure 3 and Supplementary Figure A2.
In addition, the manuscript would be enhanced if authors will include a figure presenting the results of the first part of the “3.3 Effects of LHX1 Overexpression on BHLHE40 Expression”, where they overexpressed truncated versions of the BHLHE40 transcription factor.
Response: We have presented the results on presumed dominant negative forms of BHLHE40 in Supplementary Figure A4.
Lastly, authors should discuss further the implications of the discovery of this factor and its expression in the developing eye and give a future perspective about how would these discoveries advance our understanding of the specification of the vertebrate retina and RPE.
Response: Thank you for your suggestion. We have added the final paragraph in Discussion to describe the implications of the discovery of this factor (page 11, lines 437-443). We were careful not to speculate too much.
Minor points:
Line 42-43: There is no mention of bipolar cells and Muller glia in the initial description of the retina, which provides an incomplete description of the structure.
Response: We have revised the manuscript to include the two retinal cell classes (page 1, lines 44-45).
Line 58: “Strikingly, in mammals, embryonic stem cells can self-organize” – using an expression such as “embryonic stem cells from mammals” would provide more accuracy about these protocols.
Response: According to the reviewer’s comment, we have revised the manuscript (page 2, line 60).
Line 112: “sequences were checked by Sanger sequencing” – using “sequences were confirmed by Sanger sequencing” would be preferably in scientific writing.
Response: According to the reviewer’s comment, we have revised the manuscript (page 3, line 125).
Line 138: “In Ovo Electroporation” – the word “in ovo” should be italic font, and not “electroporation”.
Response: We agree that in ovo should be written in italic and not for electroporation. We just follow the journal style. We understood letters in Roman should be italicized and those in italic should be Roman on the subtitle line.
250-251: “These data indicate that BHLHE40 can drive early RPE development and repress the developmental default state of NR” – to support this data, embryos should be allowed to develop past E5, where the RPE is pigmented and specified.
Response: Under our present experimental conditions, embryos after BHLHE40 overexpression could not survive until E5. Therefore, we would like to tone down the expression as follows: These data suggest that BHLHE40 could maintain OTX2 expression in the optic vesicle, drive early RPE development and repress the developmental program for NR (page 8, lines 299-301).
275 – “After LHX1overexpression” – a space is needed between LHX1 and overexpression
Response: According to the reviewer’s comment, we have revised the manuscript (page 8, line 329).
Figure 3 A, B, C, D, E, F seem to be imaged at a different scale. Embryo number should be labeled in the figure not only on text.
Response: In the revised Figure 3 legend, scale bars have been more accurately described (page 8, lines 293-294). Embryo numbers have been described in the legend (page 7, lines 275, 278-279 and 286).
Figures 3 and 4 are missing scale bars.
Response: Scale bars have been included in both Figures and described (page 8, lines 293-294, page 9, lines 337-338).
Figure 4 – E, F, G, E’ don’t show nuclear marker, unlike H
Response: As shown in the revised Figure 4, cell nuclei in all panels were stained with nuclear fast red.

Reviewer 3 Report
In this manuscript, Kinuhata et al. provided a comprehensive study on the bHLH-containing gene BHLHE40. This study revealed that BHLHE40 is expressed in developing chicken eye primordia, and further investigation confirmed the repression effect of LHX1 to BHLH40. Overall, this manuscript shows the expression and regulation of BHLHE40 in the chicken embryo, which serves as extra evidence of bHLH family protein function, besides previously studied zebrafish. I have a few comments on this work:
1, As a bHLH transcriptional factor, BHLHE40 also binds with the E box motifs. Have the authors confirmed the protein structure and function after adding the HA-tag? Does HA-tagged BHLHE40 show the same expression profile and binding ability?
2, The deletion of the basic region of BHLHE40 is very interesting. In specific, how does this deletion alter the protein stability and function? Will this result in a more fragile version of the protein so it cannot perform any potential functions?
3, The downstream genes of the transcription factor BHLHE40 is very intriguing. Have the authors done thorough research on the known downstream genes of it? Can we learn more from the studies in other species? I would anticipate more discussion on how BHLHE40 act on the target genes.
This research is of good quality in terms of the data and manuscript structure. I look forward to the revised manuscript.
Author Response
Responses to “Comments and Suggestions for Authors by Reviewer 3”
In this manuscript, Kinuhata et al. provided a comprehensive study on the bHLH-containing gene BHLHE40. This study revealed that BHLHE40 is expressed in developing chicken eye primordia, and further investigation confirmed the repression effect of LHX1 to BHLH40. Overall, this manuscript shows the expression and regulation of BHLHE40 in the chicken embryo, which serves as extra evidence of bHLH family protein function, besides previously studied zebrafish. I have a few comments on this work:
1, As a bHLH transcriptional factor, BHLHE40 also binds with the E box motifs. Have the authors confirmed the protein structure and function after adding the HA-tag? Does HA-tagged BHLHE40 show the same expression profile and binding ability?
Response 1: Since the structure of the C-terminus is distant from the DNA-binding domain, we think that there are no adverse effect of the tagging. In fact, there is a report (listing below) in which the FLAG-tagged BHLHE40 construct was used to examine its function by overexpression and we followed them using a HA-tag instead.
Reference:
Shen L et al. J Biol Chem (2014) 289, 34, 23332–23342. doi: 10.1074/jbc.M113.526343
2, The deletion of the basic region of BHLHE40 is very interesting. In specific, how does this deletion alter the protein stability and function? Will this result in a more fragile version of the protein so it cannot perform any potential functions?
Response 2: Western blot analysis for truncated versions of BHLHE40 has been shown in the new Supplementary Figure A4. Those proteins are not fragile. Immunostaining on sections after overexpression of the HA-tagged Δbasic BHLBE40 with or without acidic extension shows stable expression of these truncated proteins.
3, The downstream genes of the transcription factor BHLHE40 is very intriguing. Have the authors done thorough research on the known downstream genes of it? Can we learn more from the studies in other species? I would anticipate more discussion on how BHLHE40 act on the target genes.
This research is of good quality in terms of the data and manuscript structure. I look forward to the revised manuscript.
Response 3: We admit its importance, but we have not yet done through research on the downstream genes. But we have included a new paragraph on the target genes in Discussion (page 10, lines 368-374).

Round 2
Reviewer 1 Report
The author's revisions are satisfactory.